# Metabolomic Profiling of the Immune Stimulatory Effect of Eicosenoids on PMA-Differentiated THP-1 Cells

**DOI:** 10.3390/vaccines7040142

**Published:** 2019-10-09

**Authors:** Abdulmalik M. Alqarni, Tharushi Dissanayake, David J. Nelson, John A. Parkinson, Mark J. Dufton, Valerie A. Ferro, David G. Watson

**Affiliations:** 1Strathclyde Institute of Pharmacy and Biomedical Sciences, University of Strathclyde, 161 Cathedral Street, Glasgow G4 0RE, UK; v.a.ferro@strath.ac.uk (V.A.F.); d.g.watson@strath.ac.uk (D.G.W.); 2Department of Pharmaceutical Chemistry, College of Clinical Pharmacy, Imam Abdulrahman Bin Faisal University (University of Dammam), Dammam 31441, Saudi Arabia; 3WestCHEM Department of Pure and Applied Chemistry, University of Strathclyde, 295 Cathedral Street, Glasgow G1 1XL, UK; tharushi.dissanayake.2014@uni.strath.ac.uk (T.D.); david.nelson@strath.ac.uk (D.J.N.); john.parkinson@strath.ac.uk (J.A.P.); mark.dufton@strath.ac.uk (M.J.D.)

**Keywords:** (Z)-11-eicosenol, methyl cis-11-eicosenoate, cis-11-eicosenoic acid, pro- and anti-inflammatory cytokines, LPS stimulation, THP-1 cells, macrophages, adjuvant vaccine

## Abstract

Honey bee venom has been established to have significant effect in immunotherapy. In the present study, (Z)-11-eicosenol-a major constituent of bee venom, along with its derivations methyl cis-11-eicosenoate and cis-11-eicosenoic acid, were synthesised to investigate their immune stimulatory effect and possible use as vaccine adjuvants. Stimuli that prime and activate the immune system have exerted profound effects on immune cells, particularly macrophages; however, the effectiveness of bee venom constituents as immune stimulants has not yet been established. Here, the abilities of these compounds to act as pro-inflammatory stimuli were assessed, either alone or in combination with lipopolysaccharide (LPS), by examining the secretion of tumour necrosis factor-α (TNF-α) and the cytokines interleukin-1β (IL-1β), IL-6 and IL-10 by THP-1 macrophages. The compounds clearly increased the levels of IL-1β and decreased IL-10, whereas a decrease in IL-6 levels suggested a complex mechanism of action. A more in-depth profile of macrophage behaviour was therefore obtained by comprehensive untargeted metabolic profiling of the cells using liquid chromatography mass spectrometry (LC-MS) to confirm the ability of the eicosanoids to trigger the immune system. The level of 358 polar and 315 non-polar metabolites were changed significantly (*p* < 0.05) by all treatments. The LPS-stimulated production of most of the inflammatory metabolite biomarkers in glycolysis, the tricarboxylic acid (TCA) cycle, the pentose phosphate pathway, purine, pyrimidine and fatty acids metabolism were significantly enhanced by all three compounds, and particularly by methyl cis-11-eicosenoate and cis-11-eicosenoic acid. These findings support the proposed actions of (Z)-11-eicosenol, methyl cis-11-eicosenoate and cis-11-eicosenoic acid as immune system stimulators.

## 1. Introduction

Adjuvants, in the context of vaccines, are described as substances capable of enhancing and modulating antigen-specific immune responses to improve vaccine efficacy [1]. The immune stimulating effects of adjuvants was first established with the addition of aluminium potassium sulphate or aluminium salts to human vaccines [2,3]. Today, a better understanding of immune responses has increased the availability and variety of vaccine adjuvants to include virosomes, MF59 and AS04 [3]. Adjuvants are now in widespread use, but their development has been empirical, without a clear understanding of their molecular and cellular mechanisms of action. However, several studies have suggested that adjuvants act by enhancing T and B cell responses, by stimulating innate immunity and by increasing the magnitude of the adaptive responses to the vaccine [4,5,6]. Adjuvants also stimulate a strong and comprehensive immune response to antigen by mimicking natural defensive trigger molecules, such as endogenous immune-active substances (e.g., chemokines and cytokines) or other natural compounds (e.g., vitamin E and saponins) [3].

A crucial need exists for the development and investigation of new vaccine adjuvant compounds that have the ability to induce immune responses. However, the molecular and cellular mechanisms that regulate the activity of the majority of human-licensed adjuvants are still only partially characterised. The adjuvant derivatives of bacterial components, such as lipopolysaccharide (LPS), cytidine–phosphate–guanosine (CpG) oligonucleotides and monophosphoryl lipid A (MPL), are probably the most characterised at present [7]. MPL works as an agonist of Toll-like receptor (TLR4), which is expressed on antigen presenting cells (APCs), such as dendritic cells (DCs) and macrophages, and promotes cytokine expression, antigen presentation and migration of the APCs to the T-cell area [8,9]. These compounds act as microbial sensors called pathogen-associated molecular patterns (PAMPs), and they activate pattern recognition receptors (PRR) and subsequently TLR [10].

LPS is the most widely studied of the TLR4 ligands, and its adjuvant derivatives (which are less toxic) have been the basis of virtually all clinical trials of adjuvant TLR4 agonists [11]. The interactions of LPS derivatives with TLR4 are restricted mainly to their lipid A portion, which are composed of polyacylated diglucosamine lipids [12]. A combination of specific adjuvants with TLR agonists has been proposed to optimise vaccines [13]. Therefore, evaluation of the LPS enhancement of cytokine productions and a metabolic interpretation of LPS immune-modulatory effects would be a promising strategy for investigating proposed new adjuvants. 

Honey bee (*Apis mellifera*) venom is a very complex mixture of substances, including organics, peptides and enzymes. These ingredients contribute a variety of biological activities towards the overall toxic shock [14], and have attracted attention as possible leads for drug discovery. Some components have been applied to the treatment of inflammation and cancer, for example, [15,16], but their use as vaccine adjuvants is still under investigation. Melittin, a major lytic peptide found in bee venom, has been proposed as a vaccine adjuvant due to its confirmed ability to enhance TNF-α, IL-1β and IL-6 cytokine production within the THP-1 monocyte-derived macrophage cell line [17,18].

An organic compound of interest in honey bee venom is (Z)-11-eicosen-1-ol. This is a major component of the alarm pheromone mixture co-secreted with the venom [19]. Insect pheromones are typically volatile organics that are released to warn of danger [20]. (Z)-11-eicosen-1-ol prolongs the effectiveness of isopentyl acetate, another key component of the alarm pheromone secretion, thereby increasing the speed of the aggressive response [14,19]. A similar compound, (Z)-9-eicosen-1-ol, was detected in *A. mellifera* venom, resembling the compound described by Pickett et al. and differing only with respect to the double bond position [18]. It was found to enhance the release of pro-inflammatory cytokines, other than IL-6 [18].

Immune metabolism is a rapidly growing area of research. Recently, several studies have shown the importance of metabolites, such as succinate and itaconate, in the modulation of the innate immune response of macrophages [21,22]. The exposure of macrophages to various immune stimuli, including pathogenic antigens and cytokines, initiates several signaling cascades by interactions with receptors (i.e., cytokine receptors or PRR) [23]. This intracellular and extracellular signaling is expected to elicit major changes in metabolites to promote the required alteration of the cell phenotype and changes in anabolic and catabolic pathways [23]. Immune cells use different metabolic pathways to provide adequate energy generation and cell survival during cell growth and proliferation. These metabolic pathways, which include fatty acid synthesis, rely on products from other pathways, such as glycolysis and the tricarboxylic acid (TCA) cycle, to provide key synthetic precursors [24]. For this reason, metabolomics analysis of immune cells can provide a more complete understanding of the physiological state of an organism and of the alterations occurring in the metabolome, as metabolites are often considered the end stage of biological processes.

A growing number of findings highlight the crucial role of metabolic reprogramming in macrophage activation. Immune cells utilise five metabolic pathways: glycolysis, the TCA cycle, the pentose phosphate pathway (PPP), fatty acid synthesis and amino acid metabolism [24]. The present study investigated the metabolic responses of phorbol 12-myristate 13-acetate (PMA)-differentiated THP-1 cells treated with three different forms of synthetic honey bee venom eicosenoid compounds: (Z)-11-eicosenol, methyl cis-11-eicosenoate and cis-11-eicosenoic acid. The overall goal was to determine how these compounds might trigger an immune response and further used as vaccine adjuvants. Comprehensive metabolic profiling and assessments of pro- and anti-inflammatory cytokines (TNF-α, IL-1β, IL-6 and IL-10) were also conducted following exposure of LPS-stimulated THP-1 macrophage cells with these compounds. The observation of a synergistic effect of the investigated compounds with LPS in enhancement of immune stimulatory activation and cytokine production suggested the potential use of honey bee venom components as immune-modulatory agents and as vaccine adjuvants, which support their use as pro- rather than anti-inflammatory agents. 

## 2. Materials and Methods

### 2.1. Sample Preparation

Methyl *cis*-11-eicosenoate (CAS 2390-09-2) and *cis*-11-eicosenoic acid (CAS 5561-99-9) were purchased from Sigma-Aldrich and used directly.

*Cis* (Z)-11-eicosenol (CAS 62442-62-0) was obtained by selective reduction of methyl *cis*-11-eicosenoate with lithium aluminium hydride. Briefly, dry diethyl ether was added to the hydride and stirred in an ice bath for 30 min. The ester was added dropwise, stirred for 30 min, then refluxed gently for 60 min. The reaction mixture was worked up by quenching with wet ether and washing with potassium sodium tartrate solution and water. The extracted organic layer was dried over sodium sulphate, filtered and excess solvent evaporated with an air stream to obtain the product. Confirmation of the reduction was obtained via IR and the full structure of the intended product was confirmed by NMR, including by 1D ^1^H and ^13^C-{^1^H} NMR spectra and 2D [^1^H, ^13^C]-HSQC and 2D [^1^H, ^1^H] DQFCOSY and TOCSY NMR spectra Data and their assignments were confirmed directly with that of the natural product as previously assigned [18].

^1^H NMR (1D, 600 MHz, DMSO-*d*_6_): δ_H_ 0.84–0.86 (t, *J* = 13 Hz, 3H, C1), 1.24–1.30 (m, 26H, C2/C3/C4/C5/C6/C7/C12/C13/C14/C15/C16/C17/C18), 1.35–1.40 (m, 2H, C19) 1.96–1.99 (q, *J* = 19 Hz, 4H, C8/C11), 3.35–3.38 (m, 2H, C20), 4.30–4.31 (m, 1H, OH), 5.29–5.34 (sept, 2H, C9/10).

^13^C-{^1^H} NMR (1D, 150 MHz, DMSO- *d*_6_): δ 13.90 (C1), 22.07 (C2), 25.50 (C18), 26.53 (C8/C11), 28.56 (C5), 28.67 (C6), 28.81 (C7), 28.85 (C12), 28.95 (C14), 29.00 (C15), 29.07 (C16), 29.08 (C17), 31.28 (C3), 32.54 (C19), 60.70 (C20), 129.61 (C9/C10).

### 2.2. Cell Culture and Differentiation

THP-1 cells (American Type Culture Collection, ATCC^®^, Manassas, VA, USA) were cultured to a seeding density of 1 × 10^5^ cells/mL in Roswell Park Memorial Institute (RPMI) medium 1640 (Thermo Fisher Scientific, Loughborough, UK) with addition of L-glutamine (2 mmol/L, Life Tech, Paisley, UK), foetal calf serum (FCS, 10% *v*/*v*, Life Tech) and penicillin/streptomycin (100 IU/100 µg/mL, Life Tech). Sub-cultures were prepared every 2 to 4 days in fresh media and incubated at 37 °C and 100% humidity in 5% CO_2_ prior to differentiation by addition of phorbol 12-myristate 13-acetate (PMA, 60 ng/mL final concentration, Sigma-Aldrich, Dorset, UK). The cells were then incubated for 48 h, after which the PMA-containing media were replaced with fresh media and the cells rested for an additional 24 h prior to examination by light microscopy.

### 2.3. Cell Viability Assay

THP-1 cells were cultured in 96-well plates at a seeding density of 1 × 10^5^ cells/mL. The cells were incubated for 24 h at 37 °C and 100% humidity in 5% CO_2_ prior to treatment with eicosenoid compounds at various concentrations (1.2 to 150 µg/mL) followed by a further 24 h incubation. Controls included cells alone (no treatment), medium alone (background) and dimethyl sulphoxide (1.5% *v*/*v* DMSO, positive) were added. The plates were then incubated for an additional 24 h following addition of resazurin salt solution (0.1 mg/mL, 10% *v*/*v* final concentration) before examining the fluorescence at λEx = 560 nm and λEm = 590 nm (SpectraMax M5, Molecular Devices, Sunnyvale, CA, USA). A background correction was performed and the call viability for each treatment concentration was determined with respect to the mean viability of the negative control (*n* = 3). Mean inhibitory concentration (IC50) values and dose–response curves were obtained using GraphPad Prism for Windows v 5.00 (GraphPad Software, San Diego, CA, USA).

### 2.4. Cytokine Production

The differentiated cells were incubated for a further 24 h in the presence and absence of LPS (0.5 µg/mL, Sigma-Aldrich) with the final eicosenoid concentrations presented in Table 1. The prepared medium was obtained and frozen until needed for the enzyme-linked immunosorbent assay (ELISA) (*n* = 3).

### 2.5. Enzyme-Linked Immunosorbent Assay (ELISA)

The production of inflammatory cytokines TNF-α, IL-1β, IL-6 and IL-10 were quantified using ELISA Ready-Set-Go kits (Thermo Fisher Scientific, Bremen, Germany) in accordance with the supplier’s protocol. Sulphuric acid solution (2N) was added to stop the reaction prior to reading the absorbance at 560 nm (SpectraMax M5, Molecular Devices) and subtracting the absorbance at 570 nm.

### 2.6. Metabolite Extraction

THP-1 cells differentiated with PMA were cultured in 6-well plates at a seeding density of 4.5 × 10^5^ cells/mL (*n* = 6) for 48 h, after which the medium was removed by aspiration and replaced prior to incubation with LPS (0.5 µg/mL) either alone or together with (Z)-11-eicosenol (9 µg/mL), eicosenoate (150 µg/mL) and eicosenoic acid (40 µg/mL) for another 24 h. Extraction of the metabolites was carried out as described previously [17]. Seven different analytical standard solutions were prepared by adding each metabolite standard (10 µg/mL final concentration) containing glycine-^13^C_2_ [25]. Combined quality control (QC) samples were prepared by pipetting 20 µL from each individual sample, mixing and transferring to a high-performance liquid chromatography (HPLC) vial.

### 2.7. LC-MS Conditions

LC-MS was carried out using an Exactive Orbitrap (Thermo Fisher Scientific, Bremen, Germany); the conditions used were as described in our previous paper [17].

### 2.8. Data Extraction and Statistical Analysis

Data extraction was performed using the MZMatch software (SourceForge, La Jolla, CA, USA) and the metabolite peaks were filtered, compared and characterised using the Microsoft Excel IDEOM [26] as described in our previous paper [17]. The metabolite data were also subjected to multivariate analysis by fitting PCA-X and OPLS-DA using the SIMCA-P software v.14.0 (Umetrics, Umea, Sweden). Univariate comparisons were performed using Microsoft Excel and paired t-tests between treated and control cells and differences were considered significant at *p* < 0.05. Using ELISA, standard calibration curves were plotted by fitting the average optical density (OD) values of TNF-α, IL-1β, IL-6 and IL-10 to 4-parameter logistic (4-PL) regression curves. Each standard concentration assayed in duplicate (*n* = 2), as shown in Appendix A.

## 3. Results

### 3.1. Cytotoxicity of Eicosenoid Compounds against PMA-Differentiated THP-1 Cells 

The potential cytotoxicity of (Z)-11-eicosenol (11E-OH), eicosenoate (11E-ester) and eicosenoic acid (11E-acid) (Figure 1) on PMA-differentiated THP-1 cells was evaluated to select an appropriate final concentration for further tests. Clear dose-dependent toxicity to THP-1 cells was observed for 11E-OH and its 11E-acid form. The lowest IC50 value at 19.88 µg/mL was observed for 11E-OH, whereas this value was 90.98 µg/mL for 11E-acid (Figure 2A–C). By contrast, the 11E-ester form was nontoxic to THP-1 cells, with an IC_50_ value greater than 150 µg/mL. ELISAs were also conducted to assess the cytokine levels in THP-1 derived-macrophage cells upon treatment with these three compounds. As shown in Table 1, the final concentrations for cytokine assessments were chosen as those that were below the IC_50_ values and resulted in >90% of the cells remaining viable. There was no difference in effect between the solvent control DMSO alone (1.5% final concentration) and negative control media on cell viability.

### 3.2. Effect of Eicosenoid Compounds on Pro-Inflammatory TNF-α Cytokine Production

Using ELISA, the effects of the eicosenoid compounds on the production of TNF-α cytokine are shown in Figure 3 and Appendix A. The levels of secreted TNF-α by PMA-differentiated THP-1 cells were slightly or negligibly affected by all three forms of eicosanoid when combined with LPS. The increase was only statistically significant (*p* < 0.05) with 11E-acid when compared with LPS alone. (Z)-11-eicosenol compounds on their own significantly enhanced the production of TNF-α when compared with untreated cells.

### 3.3. Effect of Eicosenoid Compounds on Pro-Inflammatory IL-1β Cytokine Production

Compared to TNF-α, the enhancement in the production of IL-1β by (Z)-11-eicosenol, eicosenoate and eicosenoic acid in LPS co-stimulated THP-1 cells was much more pronounced. The release of IL-1β was greatly enhanced (ratio > 1.0) by approximately 84% and was statistically significant when compared with LPS alone. (Z)-11-eicosenol (~50%) enhanced the production of this cytokine upon stimulation with LPS, although the increase was not significant when compared with LPS alone (Figure 4 and Appendix A). The level of IL-1β was enhanced by (Z)-11-eicosenol and eicosenoic acid alone, in the absence of LPS, when compared with the untreated control; however, the effects were not statistically significant.

### 3.4. Effect of Eicosenoid Compounds on Pro-Inflammatory IL-6 Cytokine Production

The levels of IL-6 cytokine were evaluated to confirm the previously reported decrease in response to eicosenoid compounds [18]. A decrease in IL-6 levels in THP-1 derived macrophage cells was also observed in the present study in response to all three forms of synthetically prepared eicosenoids. The levels of this cytokine were significantly decreased (*p* < 0.05) when compared with LPS alone. Surprisingly, no detectable amount of IL-6 was produced by the cells treated with eicosenoate (11E-ester) in the presence of LPS (Figure 5). Unstimulated macrophage-like THP-1 cells did not produce any IL-6. The same was true for cells stimulated with eicosenoids alone (Appendix A).

### 3.5. Effect of Eicosenoid Compounds on Anti-Inflammatory Il-10 Cytokine Production

A decrease in the levels of the anti-inflammatory IL-10 cytokine would support the use of the eicosenoid compounds as immune response stimulators. Combination treatments with LPS significantly decreased the level of this cytokine in PMA-differentiated THP-1 cells when compared with LPS alone. The extent of the reductions in the release of IL-10 were about 30%, 80% and 50% in response to (Z)-11-eicosenol, eicosenoate and eicosenoic acid treatments, respectively, when compared to treatment with LPS alone (Figure 6 and Appendix A).

### 3.6. Effect of Eicosenoid Compounds on Polar THP-1 Cell Metabolites

Untargeted metabolic profiling of PMA-differentiated THP-1 cells was performed using LC-MS analysis. Samples were prepared by incubation of the macrophage cells with LPS and one of the three forms of the eicosenoids and compared to control untreated cells (C). Multivariate and univariate statistical analysis were used to visualise and examine the metabolite effects on the following treatment combinations: T1 (9 µg/mL (Z)-11-eicosenol + 0.5 µg/mL LPS), T2 (150 µg/mL eicosenoate + 0.5 µg/mL LPS) and T3 (40 µg/mL eicosenoic acid + 0.5 µg/mL LPS). The effect of LPS alone was also evaluated to confirm the previous findings [17] and to investigate new pathways that might be involved in immune stimulation by eicosenoids. LPS synergism was clearly evident from the cytokine assessment, particularly with IL-1β and IL-10. Further metabolic profiling of these combination treatments would aid in determining how they inhibit or stimulate the immune response.

As shown in Figure 7A, principle component analysis (PCA) showed an absence of outliers. In addition, pooled quality control samples (QC, P1–6) produced a single tight cluster in the centre of the dataset, confirming the stability, precision and validity of the instrumental analytical method. Orthogonal partial least squares discriminant analysis (OPLS-DA), a supervised model for sample classification, showed a clear separation of each combination treatment, indicating unique metabolite profiling (Figure 7B). The OPLS-DA model parameters and validation of the plot suggest a strong model, with the *p* value associated with the cross-validation (CV)-ANOVA = 1.96 × 10^−21^, indicating that the model was valid.

Pooled quality control samples were injected at intervals (*n* = 6) during the course of the run and used for further filtration of the dataset based on relative standard deviations (RSD). Metabolites with RSD values >30% within the pooled samples were excluded. The univariate analysis shown in Table 2 reveals a large number of metabolic changes resulting from application of eicosenoid treatments. Greater effects were observed with the 11E-ester and 11E-acid forms when combined with LPS and compared to untreated control cells. Increases in abundance were detected for a large number of metabolites, including arginine and proline, Krebs cycle (TCA cycle) compounds and purine metabolites, which are all critically involved in inflammatory processes and immune responses of the cells. 

### 3.7. Effect of Eicosenoid Compounds on Lipophilic Metabolites

In order to gain a comprehensive overview, further analysis was also carried out on the non-polar lipophilic metabolites in the cells using a reversed phase (RP) column. As shown in Figure 8A, PCA was employed for the 314 lipophilic compounds and shows the absence of the outliers. In addition, pooled quality control samples (QC, P1–6) clustered together which indicates the stability, precision and validity of the instrumental analytical method. OPLS-DA shows a clear separation of each combination treatment and represents a unique metabolite profile (Figure 8B). The OPLS-DA model parameters and validation of the plot suggest a strong model with the *p* CV-ANOVA = 0.0094, indicating that the model was valid (*p* < 0.5). From the data visualisation below (Figure 8B), strong effects can be predicted on the level of lipophilic metabolites when ester (T2) and acid (T3) treatments are applied to the THP-1 cells. 

Table 3 summarises the list of metabolites separated on the ACE C4 column in cells treated with eicosenoids in combination with LPS. The metabolites were identified by matching the retention times to those of a standard mixture of known fatty acids.

## 4. Discussion

A recognition of the crucial role of the regulation of the adaptive response and induction of the innate immune response has led to a reassessment of the role of adjuvants in vaccinology. Recent studies oninnate immunity activation of macrophages and DCs are now providing better glimpses into the mechanisms underlying adjuvant actions. These new insights now support the development of novel adjuvants and combinations of adjuvants to enhance the recognition of antigens by the immune system and to induce more potent cellular immune responses that exploit the advantages of each individual component. To the best of our knowledge, no study has yet investigated the eicosenoid effect on immune macrophage cells. Therefore, the aim of this study was to assess the ability of three eicosenoid derivatives to induce specific immunological functions of THP-1 macrophage cells by examining cytokine production and eicosenoid-induced alterations in the cell metabolome.

THP-1 cell viability in the presence of (Z)-11-eicosenol, methyl cis-11-eicosenoate and cis-11-eicosenoic acid revealed different IC_50_ values, indicating an effect of functional group substitution on cellular respiration. The lowest IC_50_ value was obtained with (Z)-11-eicosenol, which contains a hydroxyl group (Figure 1), in comparison with the ester and carboxylic acid forms. Several studies have assessed the effect of functional groups on different cell lines. For example, Sakagami et al. reported the structure–activity relationships of 11 piperic acid ester derivatives based on their cytotoxic effects against oral squamous cell carcinoma cell lines and found that addition of two hydroxyl groups had the highest cytotoxic effect [27]. Similarly, polyhydroxylated analogues of resveratrol, a natural polyphenol compound, showed higher cytotoxic effects [28].

Previously, (Z)-9-eicosenol was purified from honey bee venom and its stimulatory effect on TNF-α and IL-1β secretion was reported in the U937 cell line stimulated with LPS, where a surprising observation was significant inhibition of the level of IL-6 in response to this compound [18]. For this reason, in the present study, other eicosenoid derivatives were examined for their effects on the cellular immune response and in particular for their effects on the production of pro and anti- inflammatory cytokines by THP-1 macrophage cells. All three derivatives showed similar bioactivity with respect to the induction of cytokine levels, as described previously [18]. A small effect was observed for the secretion of TNF-α, whereas IL-1β production was enhanced significantly by the 11E-ester and 11E-acid forms, when combined with LPS. By contrast, the level of IL-6 decreased, suggesting the presence of a more subtle mechanism of action for eicosenoids. In addition, release of the anti-inflammatory IL-10 cytokine was largely inhibited, supporting their immune stimulatory effect.

The complexity of the IL-6 cytokine responses has been reported extensively [29]. This cytokine has dual properties pro- and anti-inflammatory, which are referred to as its classic and trans-signaling pathways [29]. IL-6 promotes a protective effect in some inflammatory diseases, such as inflammatory bone destruction and dextran sodium sulphate-induced colitis [30,31]. Several studies have also demonstrated an association between IL-6 and IL-10 and a requirement for IL-6 in IL-10 production by T cells to suppress inflammation [32,33]. Interestingly, in the current case, both IL-6 and IL-10 were decreased in the same manner (i.e., the 11E-ester form strongly decreased their secretion, with weaker decreases by the 11E-acid and still weaker effects by the 11E-OH form). Therefore, the important consequences on the therapeutic blockade of IL-6 as a treatment of chronic inflammatory diseases should be carefully considered. 

The ability of synthetic eicosenoid derivatives to induce immune responses and to alter cytokine production was investigated further by a comprehensive untargeted metabolomics assessment of PMA-stimulated THP-1 cells. Induction of the characteristic morphology of activated macrophages as a result of Toll-like receptor (TLR4) activation by LPS has been previously described [34]. Several biomarkers, including nitric oxide (NO), are used to monitor the inflammatory status of these cells.

Inducible nitric oxide synthase (iNOS) is the main enzyme responsible for the release of large amounts of NO through the conversion of L-arginine to NO and citrulline [35]. Arginine metabolism is also involved in the regulation of inflammation through its breakdown into ornithine and urea by arginase [36]. Upregulation of arginine and proline metabolism was significantly identified by metabolomic analysis in response to treatment with the ester and acid forms, as indicated by the alterations in the levels of NO-related metabolites, including L-arginine, L-citrulline, N-(L-arginino) succinate, L-proline and L-ornithine (Table 2). L-proline and L-ornithine work as precursors for the formation of pro-proliferative polyamines and production of extracellular matrix, as a repair phase response [36]. The enhancement of cytokine production by the 11E-acid form is evident mainly by the alteration of several pathways, including arginine and proline metabolism, where a significantly higher number of metabolites was altered by this treatment (Appendix A).

Several studies have shown that metabolic reprogramming of the cell regulates macrophage activation. In the present study, the findings for LPS activation of THP-1 macrophages regarding specific pathway and biomarker metabolites are consistent with those of previous reports [17,37]. In general, pro-inflammatory stimuli cause the macrophages to undergo a metabolic switch from oxidative phosphorylation (OXPHOS) to glycolysis [38]. The lactate flux from pyruvate increases in response to a reduction in acetyl-coenzyme A (acetyl-CoA) levels [38,39] leading to dysfunctional activity of the TCA cycle, which is, in turn, compensated for by an increase in glycolytic flux to ensure a rapid regeneration of adenosine triphosphate (ATP) [40].

LPS enhanced activities clearly shown by the combination treatments with eicosenoid derivatives in this study. The levels of ATP were reduced in the treatments with LPS alone and in combination with eicosenoids, when compared with untreated control cells suggesting an increased requirement in the treated cells, leading to upregulation of most of the glycolysis metabolites, including fructose 1,6-bisphosphate and fructose 6-phosphate. In order to maintain a high cellular redox state, PPP activity was increased. LPS is reported to suppress the expression of carbohydrate kinase-like protein (CARKL), which is associated with a greater flux into glycolysis and away from the non-oxidative PPP [40]. This would lead to a significant decrease of sedoheptulose-7-phosphate (S7P) [41]; however, the opposite was observed for all three treatments with LPS when compared with the untreated control. The reason for this is not clear.

The TCA cycle intermediates, such as succinate and citrate, are critical in M1 macrophage activation and are positively associated with inflammation [38]. High levels of succinate were observed in the present study in response to LPS alone and with the 11E-acid and LPS combination. This increase has been attributed to glutamine metabolism following dysfunction of the TCA cycle, which normally would provide the source for succinate generation [42]. Succinate drives inflammation through the inhibition of prolyl hydroxylase (PHD) and further stabilisation of hypoxia-inducible factor-1α (HIF-1α) [38]. LPS triggers T-cell activation and the adaptive immune system through an increased expression of the citrate transport carrier, which could lead to cytosolic citrate accumulation. This citrate is then converted to acetyl-CoA and oxaloacetate for fatty acid synthesis and generation of NO and reactive oxygen species (ROS), respectively [43]. Interestingly, in the current study, the citrate levels were approximately 20% higher in the 11E-ester and 11E-acid combinations with LPS when compared with LPS alone. Clearly, citrate and succinate are important signaling molecules in innate and adaptive immunity [44]. Furthermore, the levels of itaconate, a metabolite synthesised via decarboxylation of cis-aconitate, were elevated in the LPS-activated macrophages. This metabolite has been suggested to work as an anti-microbial, to limit inflammation and to have a crucial role in macrophage-based immune responses [45,46,47]. Interestingly, cis-aconitate was significantly elevated following the ester and acid form treatments, while itaconate was not detected (Table 2).

An imbalance between cellular oxidants and antioxidants can potentially lead to oxidative stress and cell damage. Therefore, antioxidant defence is required for cellular adaptation to stress conditions [48]. Glutathione (GSH), an important protective antioxidant tripeptide, has been implicated in inflammatory responses and immune modulation [49]. GSH is oxidised to glutathione disulphide (GSSG) when it reacts with peroxide (H_2_O_2_) in the presence of glutathione peroxidase, an enzyme that facilitates the inactivation of peroxide [50]. GSSG can be reduced again using nicotinamide adenine dinucleotide phosphate (NADPH). Therefore, regulation of both the NADPH/NADP^+^ and GSH/GSSG ratios is tightly coupled to the control of oxidative stress (Figure 9) [51,52]. A decrease in the level of GSH and an increase in its product GSSG was observed in this study; this response has been reported as a hallmark of oxidative stress [53,54]. A large depletion in the level of NADPH and an increase in NADP+ were observed in the current study after the combination treatment of eicosenoids and LPS (Table 2). NADPH generates ROS through NADPH oxidase, and it also serves as a substrate for the conversion of arginine to citrulline and NO [55,56]; arginine and citrulline were also strongly elevated in response to the treatments with LPS in combination with 11E-ester or 11E-acid (Figure 9).

Enhanced activity of the PPP boosts the production of purine and pyrimidine nucleotides for further biosynthesis in activated cells [38]. An increase in inosine, guanosine, hypoxanthine, xanthine and urate levels was detected in the present study. In purine metabolism, xanthine oxidase catalyses the oxidation of hypoxanthine to xanthine and H_2_O_2_ and then to uric acid [23]. Xanthine oxidase inhibitors, such as allopurinol, can limit the inflammation in a mouse model of arthritis [57]. This suppression of inflammation is associated with an elevation of several metabolites, including inosine, guanosine and xanthosine, which are substrates of purine nucleoside phosphorylase (PNP). A loss of PNP activity or accumulation of one or more enzyme substrates has been linked extensively to immune deficiency [58]. However, the observation of upregulation in the level of substrates and products of xanthine oxidase and PNP enzymes in the present study (Figure 10) suggests an ability of eicosenoid derivatives (particularly 11E-ester and 11E-acid form) to modulate the activity of these enzymes and to perturb the purine nucleotide pathway in order to boost the immune system through the provision of ROS.

Recent research has reflected a significantly increased interest in fatty acid metabolism and its effects on inflammation and immune cells. In activated macrophages, the importance of mitochondrial fatty acid metabolism or fatty acid oxidation (FAO) has been considered to reflect the regeneration of ATP and a compensation for the reduction in ATP levels [59]. Several studies have highlighted the possibility of attenuating inflammation by promoting macrophage FAO, which also reduces lipid-induced triglyceride accumulation [60]. Inhibition of FAO in THP-1 cells macrophages exacerbates palmitate-induced endoplasmic reticulum stress and inflammation responses [61]. In addition, LPS-induced inflammation results in elevated levels of kidney triglyceride, fatty acid and cholesteryl ester through a decrease in fatty acid beta oxidation [62]. Intriguingly, the levels of fatty acids were highly elevated in response to the treatments with LPS in combination with eicosenoids when compared with cells treated with LPS alone or untreated controls. In the present study, treatment with 11E-ester and 11E-acid in particular led to increases in a large number of fatty acids, such as arachidonic acid (i.e., eicosatetraenoic acid), docosenoic acid, tetracosenoic acid, eicosenoic acid and icosatrienoic acid (Table 3), which all are strongly correlated with inflammation and immune activation. The presence of elevated levels of several dioic acids (decandioic, dodecandioic and tetradecandoic acids) suggests that the eicosanoic acid and its ester are functioning as peroxisome proliferator ligands since dioic acids are products of peroxisome activity [63].

Lipid profiles (lipidomics) and full assessment of their levels, including eicosanoid, are effective ways to diagnose the severity and progression of several diseases [64,65]. Arachidonic acid (AA), a precursor of eicosanoids, is considered to represent a potent signal of cellular responses. AA metabolites are involved in the inflammatory and immune responses [66], and the activation of phospholipase A2 (PLA_2_) is critical for increasing the AA level and subsequent eicosanoid biosynthesis. TLR4-mediated priming has been observed to activate cytosolic calcium-dependent PLA_2_ (cPLA_2_) and to enhance cyclooxygenase (COX2) production via Nuclear factor-kappa B (NF-κB), which results in release of AA and pro-inflammatory eicosanoids [65]. In the present study, the level of AA was strongly elevated by all the combination treatments when compared to LPS alone; however, this elevation was more pronounced with 11E-ester form, which resulted in a seven-fold increase in the AA level.

Comprehensive assessment of lipid-correlated inflammatory signaling could provide a better understanding of cytokine integration with fatty acid metabolism and particular the production of eicosanoids. In general, the suppression of β-oxidation of fatty acids could result in the apparent boost in fatty acid synthesis and the reductions in ATP levels and citrate accumulation observed in the present study. A more targeted investigation is therefore required to confirm the proposed use of these eicosenoid derivatives as immune system stimulators and as vaccine adjuvants.

## 5. Conclusions

A deeper understanding of the modes of action that regulate the immune stimulatory properties of new or existing adjuvants is prerequisite for the rational design of more sophisticated vaccines. The present study identifies the main effects of synthetic compounds related to 11-eicosanol, which is found in bee venom, in modulating macrophage behaviour. In agreement with previous findings [17,37], a stimulatory effect of LPS was confirmed in PMA-differentiated THP-1 cells. Furthermore, the possible use of (Z)-11-eicosenol, methyl cis-11-eicosenoate and cis-11-eicosenoic acid to alter macrophage cytokine production was studied. Although these eicosenoid compounds enhanced the production of IL-1β and suppressed the production of IL-10, which mirrored their immune stimulatory effects, IL-6 release was inhibited, suggesting the presence of a more subtle mechanism of immune modulation. Further metabolic investigations were therefore performed to obtain a better understanding of changes in the macrophage metabolome. Alterations in metabolite levels in response to LPS treatment in the presence or absence of eicosenoids reflected the strength of the actions of each component and confirmed the potential of these compounds, particularly the ester and acid forms, to synergise LPS action. Overall, the findings supported their actions as pro- rather than anti-inflammatory agents. 

Significant alterations were observed in the metabolite levels associated with different pathways, including redox cell signaling pathways, which have been extensively associated with transcriptional immune factors and immune functions [50,67]. These alterations occurred concomitantly with upregulation of the levels of several metabolites within the arginine and proline pathways, glycolysis, the TCA cycle and purine metabolism. Moreover, marked increases occurred in the levels of several fatty acids and inflammatory biomarker metabolites, including arachidonic acid. Cellular activation and enhanced immune response were confirmed by the effects of (Z)-11-eicosenol, methyl cis-11-eicosenoate and cis-11-eicosenoic acid, the findings suggested their possible use as immune-modulating agents and vaccine adjuvants. Taken together, these findings provide a better understanding of the mechanism of immune stimulation, and they support the potential use of new adjuvants in shaping a desired immune response. Comparison with existing adjuvants would need to be carried out in order to confirm any advantages of the eicosenoids.

## Figures and Tables

**Figure 1 vaccines-07-00142-f001:**
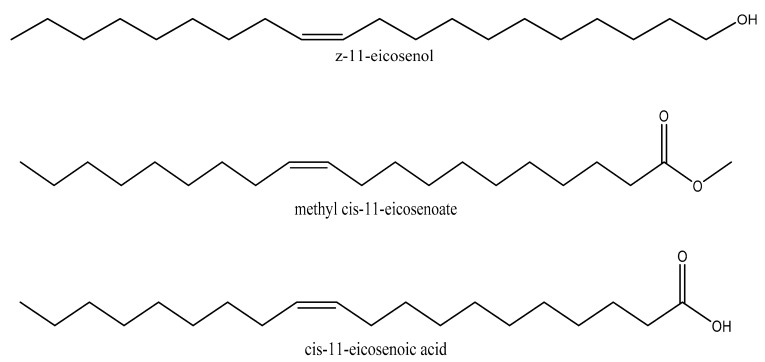
Chemical structures of (Z)-11-eicosenol, methyl cis-11-eicosenoate and cis-11-eicosenoic acid.

**Figure 2 vaccines-07-00142-f002:**
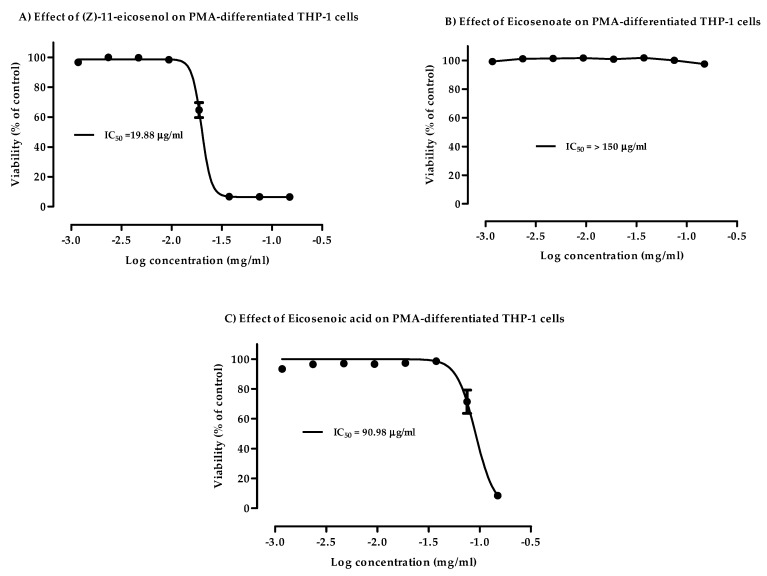
Cytotoxic effects of eicosenoid compounds at varying doses on phorbol 12-myristate 13-acetate (PMA)-differentiated THP-1 cells. (**A**) (Z)-11-eicosenol (11E-OH) compound was cytotoxic to PMA-treated cells. (**B**) Eicosenoate (11E-ester) compound was non-cytotoxic to PMA-treated cells. (**C**) Eicosenoic acid (11E-acid) compound was cytotoxic to PMA-treated cells. Each data point represents the mean ± SD (*n* = 3).

**Figure 3 vaccines-07-00142-f003:**
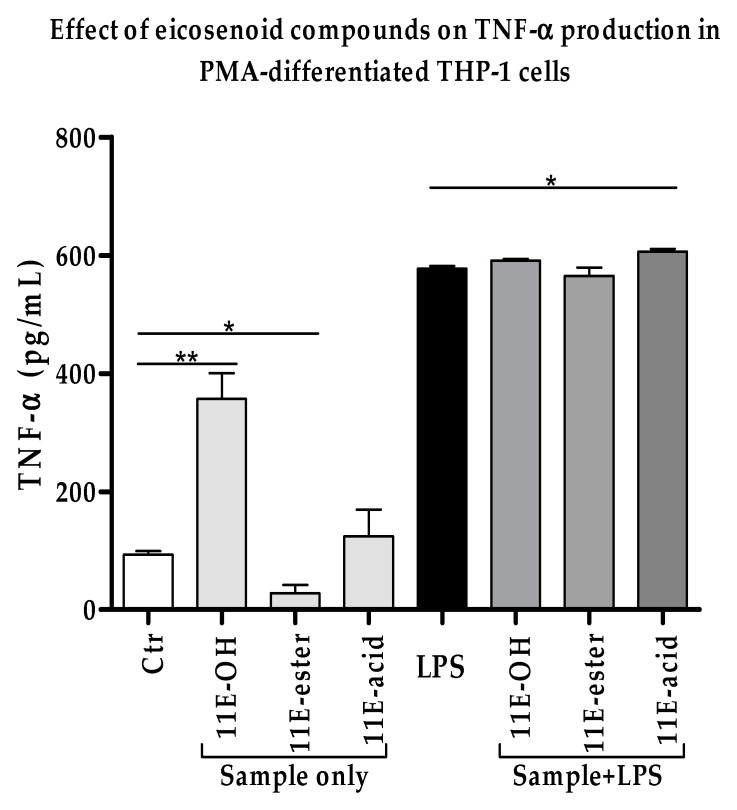
Effect of eicosenoid compounds on the production of TNF-α by PMA-differentiated THP-1 cells in the absence and presence of LPS (0.5 µg/mL). The TNF-α levels were significantly increased by 11E-acid when compared with LPS alone (*n* = 3). Ctr: Untreated control; LPS: Lipopolysaccharide; (11E-OH): (Z)-11-eicosenol; (11E-ester): Eicosenoate; (11E-acid): Eicosenoic acid; *: Significant (*p* < 0.05); **: Significant (*p* < 0.01).

**Figure 4 vaccines-07-00142-f004:**
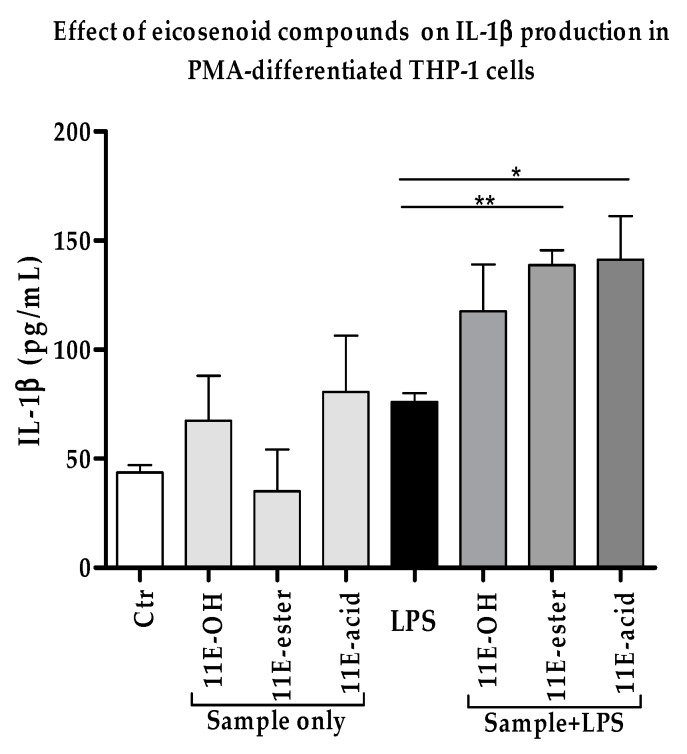
Effect of eicosenoid compounds on the production of IL-1β by phorbol 12-myristate 13-acetate (PMA)-differentiated THP-1 cells in the absence and presence of LPS (0.5 µg/mL). The IL-1β levels were significantly enhanced in 11E-ester and 11E-acid combination treatments when compared with LPS alone (*n* = 3). Ctr: Untreated control; LPS: Lipopolysaccharide; (11E-OH): (Z)-11-eicosenol; (11E-ester): Eicosenoate; (11E-acid): Eicosenoic acid; *: Significant (*p* < 0.05); **: Significant (*p* < 0.01).

**Figure 5 vaccines-07-00142-f005:**
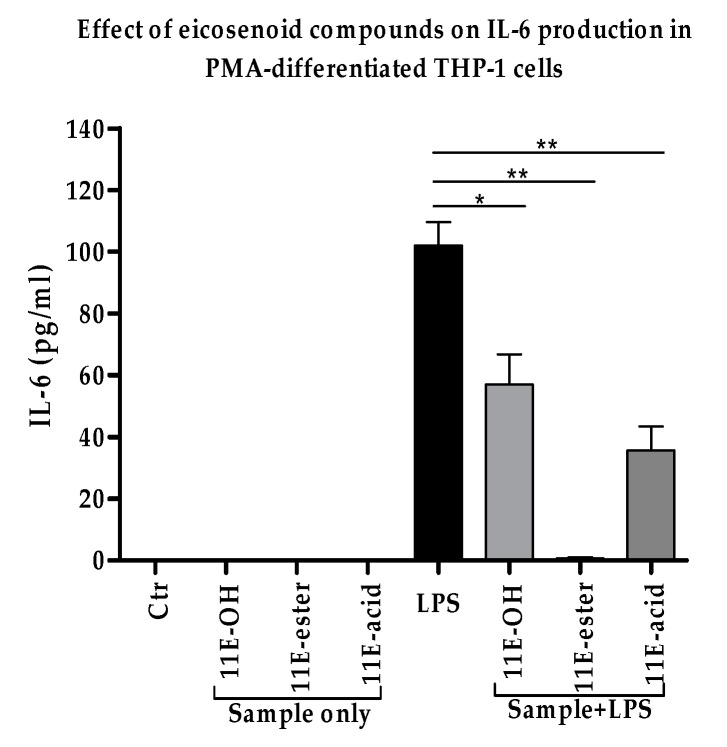
Effect of eicosenoid compounds on the production of IL-6 by phorbol 12-myristate 13-acetate (PMA)-differentiated THP-1 cells in the absence and presence of LPS (0.5 µg/mL). The IL-6 levels were significantly decreased in all three combination treatments when compared with positive control LPS (*n* = 3). Ctr: Untreated control; LPS: Lipopolysaccharide; (11E-OH): (Z)-11-eicosenol; (11E-ester): Eicosenoate; (11E-acid): Eicosenoic acid; *: Significant (*p* < 0.05); **: Significant (*p* < 0.01).

**Figure 6 vaccines-07-00142-f006:**
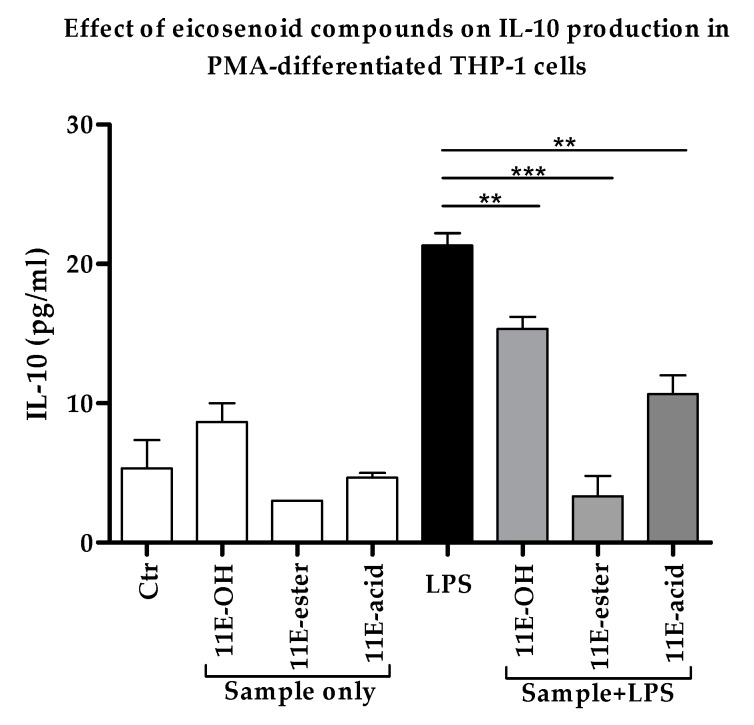
Effect of eicosenoid compounds on the production of IL-10 by phorbol 12-myristate 13-acetate (PMA)-differentiated THP-1 cells in the absence and presence of LPS (0.5 µg/mL). The IL-10 levels were significantly decreased in all three combination treatments when compared with positive control LPS (*n* = 3). Ctr: Untreated control; LPS: Lipopolysaccharide; (11E-OH): (Z)-11-eicosenol; (11E-ester): Eicosenoate; (11E-acid): Eicosenoic acid; **: Significant (*p* < 0.01); ***: Significant (*p* < 0.001).

**Figure 7 vaccines-07-00142-f007:**
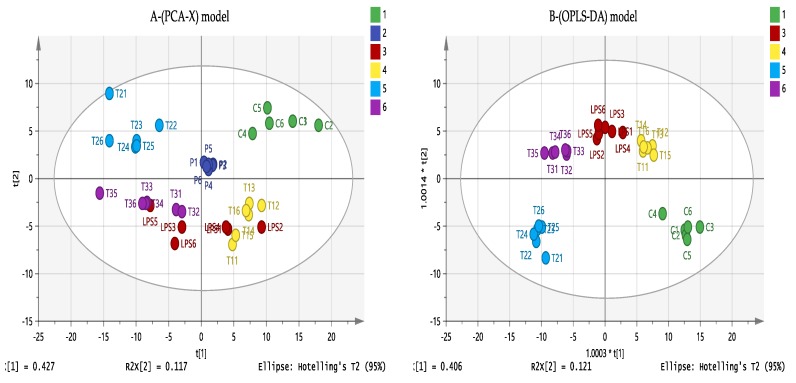
(**A**) Principle component analysis (PCA-X) vs. (**B**) Orthogonal Partial Least Squares Discriminant Analysis (OPLS-DA) score plots of THP-1 cells. The figures show a clear separation between control, pooled and treatment groups based on 358 polar metabolites separated on a ZIC-pHILIC column (*n* = 6). PCA score plot (A) gives the goodness of fit (R^2^X) = 0.807, and the goodness of prediction (Q^2^) = 0.641. OPLS-DA score plot (B) gives R^2^X = 0.829, R^2^Y = 0.962, Q^2^ = 0.874. (C: Control; LPS: Lipopolysaccharides; T1(1–6): (11E-OH) (Z)-11-eicosenol + LPS; T2(1–6): (11E-ester) Eicosenoate + LPS; T3(1–6): (11E-acid) Eicosenoic acid + LPS; *p* = pooled samples).

**Figure 8 vaccines-07-00142-f008:**
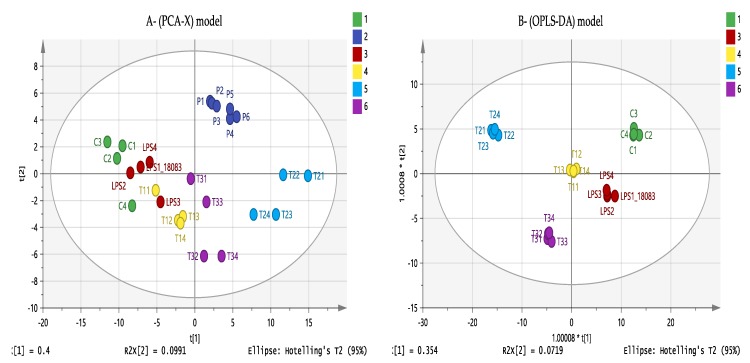
(**A**) Principle component analysis (PCA-X) vs. (**B**) Orthogonal Partial Least Squares Discriminant Analysis (OPLS-DA) score plots of THP-1 cells. The figures show a clear separation between control, pooled and treatment groups based on 314 non-polar metabolites separated on an ACE C4 column (*n* = 4). PCA score plot (A) gives the goodness of fit (R^2^X) = 0.571, and the goodness of prediction (Q^2^) = 0.403. OPLS-DA score plot (B) gives R^2^X = 0.697, R^2^Y = 0.982, Q^2^ = 0.628. (C: Control; LPS: Lipopolysaccharide; T1(1–4): (11E-OH) eicosenol + LPS; T2(1–4): (11E-ester) eicosenoate + LPS; T3(1–4): (11E-acid) eicosenoic acid + LPS; *p* = pooled samples).

**Figure 9 vaccines-07-00142-f009:**
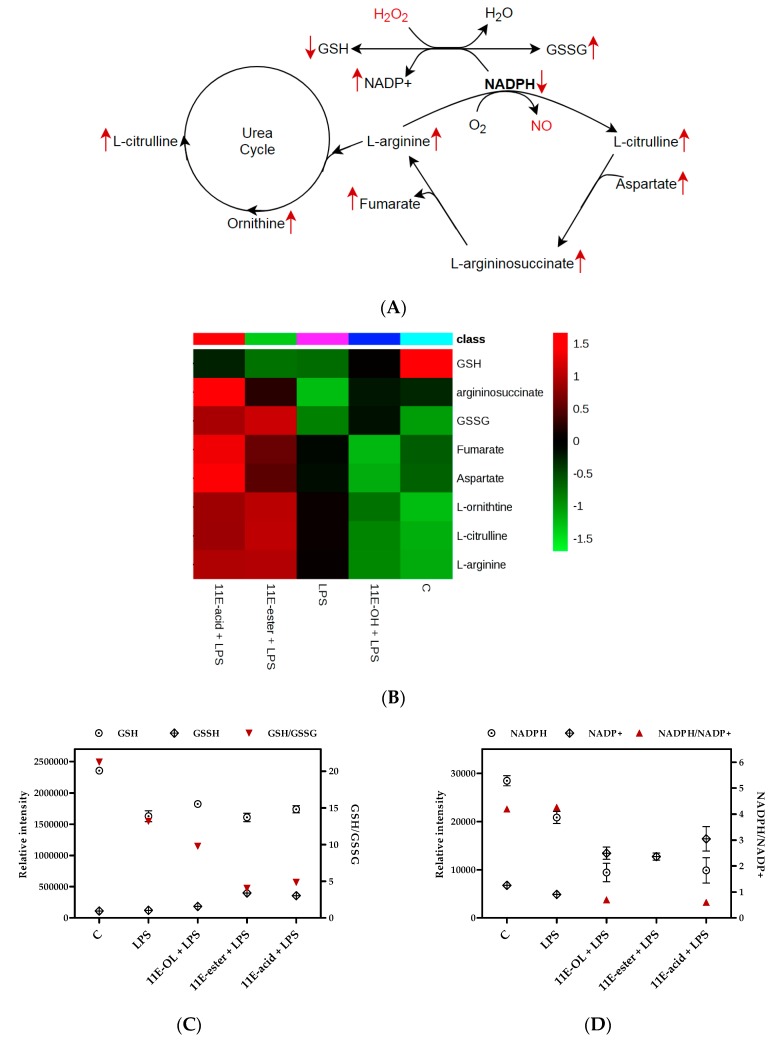
Role of nicotinamide adenine dinucleotide phosphate (NADPH) in generating hydrogen peroxide (H_2_O_2_) and nitric oxide (NO) through glutathione metabolism and arginine biosynthesis in THP-1 macrophages and show (**A**) Enhanced activity of oxidative stress related metabolites after treatment with 11E-ester+LPS and 11E-acid+LPS when compared with LPS alone. (**B**) Overexpression (dark red) of most of the significant metabolites can be observed in a heat map visualising by 11E-ester+LPS and 11E-acid+LPS treatments. (**C**) GSH, GSSG and GSH/GSSG responses after each group treatments. (**D**) NADPH, NADP+ and NADPH/NADP+ responses after each group treatments. C: Untreated control; LPS: Lipopolysaccharide; (11E-OH): (Z)-11-eicosenol; (11E-ester): Eicosenoate; (11E-acid): Eicosenoic acid; (Red rows): Response of 11E-ester+LPS and 11E-acid+LPS combination treatments.

**Figure 10 vaccines-07-00142-f010:**
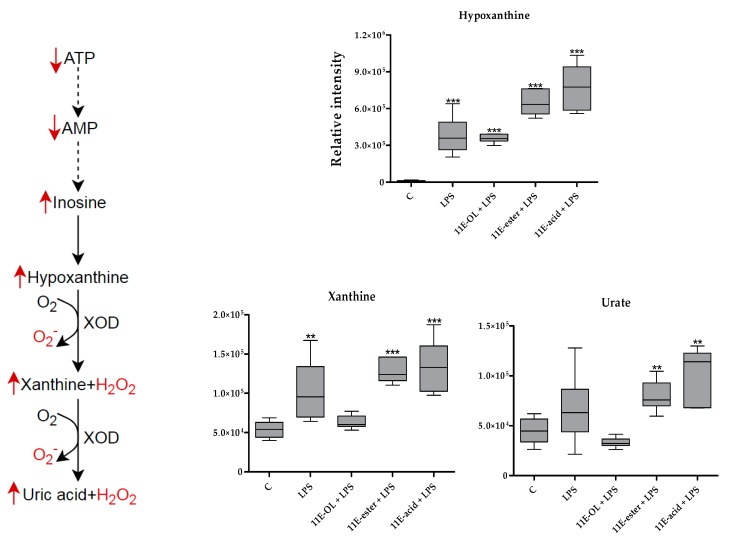
Schematic representation of the purine metabolism. This reflects how 11E-ester + LPS and 11E-acid + LPS synergise the effect of LPS alone focusing on hypoxanthine, xanthine and uric acid, which accumulated with H_2_O_2_ and superoxide (O_2_^−^) during purine degradations. C: Untreated control; LPS: Lipopolysaccharide; (11E-OH): (Z)-11-eicosenol; (11E-ester): Eicosenoate; (11E-acid): Eicosenoic acid; (Red rows): Response of 11E-ester + LPS and 11E-acid + LPS combination treatments; ATP: Adenosine triphosphate; AMP: Adenosine monophosphate; **: Significant (*p* < 0.01); ***: Significant (*p* < 0.001).

**Table 1 vaccines-07-00142-t001:** Sample groups, IC_50_ concentrations and final chosen concentrations of synthetic bee venom compounds tested in phorbol 12-myristate 13-acetate (PMA)-differentiated THP-1 cells.

Synthetic Compounds		IC50 (µg/mL)	Selected Final Concentration (µg/mL)
Group ID	Chemical Name		
**11E-OH**	(Z)-11-eicosenol	19.88	9.0
**11E-ester**	methyl cis-11-eicosenoate	>150	150
**11E-acid**	cis-11-eicosenoic acid	90.98	40.0

IC: inhibitory concentration.

**Table 2 vaccines-07-00142-t002:** Significantly changed polar metabolites in THP-1 cells treated with lipopolysaccharide (LPS), alone or in combination with one of three synthetic forms of honey bee eicosenoids (0.5 µg/mL LPS; 9 µg/mL 11E-OH; 150 µg/mL 11E-ester; 40 µg/mL 11E-acid). Data are compared with those from untreated control cells.

Mass	Rt	Putative Metabolite	LPS/C		11E-OH + LPS/C		11E-Ester + LPS/C		11E-Acid + LPS/C	
			Ratio	*p* Value	Ratio	*p* Value	Ratio	*p* Value	Ratio	*p* Value
	**Arginine and Proline Metabolism**									
129.090	15.79	4-Guanidinobutanal	1.523	0.030	1.580	0.008	0.880	ns	1.539	0.010
111.032	20.00	Pyrrole-2-carboxylate	1.200	0.012	1.384	<0.001	1.068	ns	1.260	0.003
189.064	14.09	*N*-Acetyl-l-glutamate *	0.547	0.007	0.599	0.001	1.139	ns	0.757	0.031
132.054	14.39	*N*-Carbamoylsarcosine	1.100	ns	1.051	ns	1.395	0.003	1.367	0.011
240.122	16.51	Homocarnosine	1.382	0.049	0.900	ns	1.593	<0.001	1.866	<0.001
211.036	15.28	Phosphocreatine *	0.944	ns	0.970	ns	1.723	0.001	1.285	0.024
113.059	9.94	Creatinine	1.625	ns	0.939	ns	2.104	<0.001	2.199	<0.001
130.122	26.73	Agmatine	1.509	0.021	1.041	ns	2.149	<0.001	1.884	<0.001
129.043	10.08	Oxoproline *	1.732	0.013	0.850	ns	2.331	<0.001	2.525	<0.001
175.096	16.21	L-Citrulline *	1.554	0.011	0.923	ns	2.379	<0.001	2.180	<0.001
246.133	18.71	*N2*-(d-1-Carboxyethyl)-l-arginine	2.225	0.008	1.233	ns	3.317	<0.001	3.245	<0.001
290.122	16.97	*N*-(l-Arginino)succinate	0.794	ns	1.022	ns	1.153	ns	1.596	0.015
115.063	13.07	l-Proline *	1.102	ns	0.897	ns	0.995	ns	1.208	0.036
174.112	26.73	l-Arginine *	1.593	0.022	1.089	ns	2.172	<0.001	2.160	<0.001
132.090	26.73	l-Ornithine *	1.834	0.003	1.223	ns	2.665	<0.001	2.501	<0.001
		**Glycolysis/TCA cycle**								
260.030	16.93	d-Glucose 1-phosphate *	1.231	ns	1.450	<0.001	0.833	ns	1.508	0.001
180.063	15.08	d-Glucose *	3.076	<0.001	1.901	<0.001	3.638	<0.001	3.309	<0.001
339.996	18.13	d-Fructose 1,6-bisphosphate *	1.789	<0.001	1.930	<0.001	2.217	<0.001	1.734	<0.001
260.030	16.10	d-Fructose 6-phosphate *	1.580	<0.001	1.716	<0.001	1.515	<0.001	1.653	<0.001
169.998	16.16	d-Glyceraldehyde 3-phosphate *	0.600	<0.001	0.441	<0.001	0.519	<0.001	0.404	<0.001
185.993	16.75	3-Phospho-d-glycerate	0.775	ns	1.849	ns	0.886	ns	0.934	ns
169.998	15.33	Glycerone phosphate	1.244	ns	2.368	0.014	1.804	<0.001	1.293	0.023
177.943	15.96	Pyrophosphate	0.832	0.002	0.946	ns	1.040	ns	1.015	ns
97.977	15.96	Orthophosphate	0.768	0.003	0.912	ns	1.031	ns	1.036	ns
192.027	18.13	Citrate *	1.594	<0.001	1.187	0.021	1.939	<0.001	1.904	<0.001
134.022	15.88	(S)-Malate *	1.116	ns	0.943	ns	0.869	0.002	1.124	ns
174.016	18.13	cis-Aconitate *	1.265	ns	0.984	ns	1.662	0.001	1.633	0.001
192.027	19.36	Isocitrate *	1.388	ns	1.493	ns	2.417	0.002	1.788	0.014
116.011	15.02	Fumarate *	1.186	ns	0.831	ns	1.445	0.002	1.826	<0.001
118.027	14.98	Succinate *	1.557	0.002	1.170	ns	1.346	0.005	1.819	0.001
131.070	14.97	Creatine *	1.437	0.001	1.061	ns	3.014	<0.001	1.365	0.002
809.126	12.49	Acetyl-CoA	0.730	0.006	0.907	ns	1.001	ns	0.995	ns
665.125	13.44	NADH *	0.553	<0.001	0.642	<0.001	1.330	ns	0.752	0.008
663.109	14.39	NAD+ *	0.490	<0.001	0.499	<0.001	0.683	<0.001	0.548	<0.001
506.995	16.67	ATP *	0.689	0.002	0.769	0.001	0.584	<0.001	0.724	0.001
427.030	15.30	ADP *	0.598	<0.001	0.671	<0.001	0.638	<0.001	0.762	<0.001
443.024	18.01	GDP *	0.772	0.011	0.978	ns	1.004	ns	0.931	ns
522.990	19.36	GTP *	0.975	ns	1.385	0.003	1.297	0.002	1.051	ns
	**Oxidative Stress/Pentose Phosphate Pathway**									
370.007	18.36	d-Sedoheptulose 1,7-bisphosphate	1.475	0.002	1.817	<0.001	1.565	<0.001	1.513	<0.001
232.035	15.75	d-Ribitol 5-phosphate *	0.683	0.007	0.744	ns	0.563	0.002	0.862	0.027
276.025	17.73	6-Phospho-d-gluconate *	1.606	0.006	1.480	0.004	0.568	0.001	0.921	ns
196.058	13.26	d-Gluconic acid *	0.886	ns	0.800	<0.001	0.795	0.002	0.925	ns
150.053	13.64	d-Ribose	1.307	0.017	0.899	ns	1.286	0.018	1.289	0.011
290.040	16.33	d-Sedoheptulose 7-phosphate	1.481	0.001	2.016	<0.001	1.654	<0.001	1.884	<0.001
230.019	15.35	d-Ribulose 5-phosphate	1.566	ns	4.634	0.030	1.410	0.010	1.438	0.007
230.019	15.75	d-Ribose 5-phosphate *	1.525	<0.001	1.516	<0.001	1.796	<0.001	1.230	<0.001
745.091	17.14	NADPH *	0.733	0.002	0.331	<0.001	00.00	n/a	0.347	0.004
743.076	16.87	NADP+ *	0.722	0.024	1.983	<0.001	1.879	<0.001	2.419	0.001
152.068	13.11	Xylitol *	1.423	0.001	1.063	ns	0.907	ns	1.189	0.049
196.058	13.89	d-Mannonate	2.006	ns	0.860	ns	2.406	<0.001	2.454	<0.001
166.048	13.43	d-Xylonate	1.673	0.019	0.929	ns	3.156	<0.001	2.964	<0.001
150.053	13.64	d-Ribose	1.307	0.017	0.899	ns	1.286	0.018	1.289	0.011
307.084	14.37	Glutathione	0.691	<0.001	0.774	<0.001	0.683	<0.001	0.738	<0.001
612.152	17.52	Glutathione disulfide *	1.113	ns	1.679	0.003	3.605	<0.001	3.221	<0.001
		**Purine Metabolism**								
363.058	19.36	Guanosine 3’-phosphate	0.909	ns	1.551	0.005	1.406	0.001	1.043	ns
168.028	12.41	Urate *	1.487	ns	0.738	ns	1.778	0.003	2.285	0.001
268.081	11.11	Inosine *	1.804	0.004	5.853	<0.001	12.553	<0.001	9.126	<0.001
283.092	12.83	Guanosine *	1.454	ns	4.991	<0.001	16.908	<0.001	4.976	<0.001
136.038	10.39	Hypoxanthine *	33.258	<0.001	31.216	<0.001	56.447	<0.001	67.586	<0.001
284.075	12.01	Xanthosine *	1.564	0.004	1.306	0.020	1.651	0.003	1.825	0.001
152.033	11.31	Xanthine *	1.907	0.005	1.171	ns	2.376	<0.001	2.492	<0.001
363.058	16.78	GMP *	1.016	ns	1.100	ns	0.809	ns	1.201	0.006
363.058	19.36	Guanosine 3’-phosphate	0.909	ns	1.551	0.005	1.406	0.001	1.043	ns
283.092	12.83	Guanosine *	1.454	ns	4.991	<0.001	16.908	<0.001	4.976	<0.001
347.063	13.89	AMP *	0.656	<0.001	0.660	0.001	0.326	<0.001	0.680	<0.001
		**Pyrimidine Metabolism**								
483.968	17.92	UTP *	0.355	<0.001	0.440	<0.001	0.393	<0.001	0.417	<0.001
308.041	13.81	dUMP	0.357	<0.001	0.394	<0.001	0.548	<0.001	0.394	<0.001
482.984	18.48	CTP *	0.473	<0.001	0.596	<0.001	0.688	<0.001	0.492	<0.001
403.018	17.14	CDP *	0.363	<0.001	0.440	<0.001	0.822	ns	0.541	<0.001
323.052	15.35	CMP *	0.763	0.001	0.882	0.040	1.148	ns	0.999	ns
128.058	10.57	5,6-Dihydrothymine	1.664	ns	0.698	0.006	1.395	0.007	2.075	0.033
126.043	15.31	Thymine	1.439	0.031	1.125	ns	1.995	<0.001	1.893	0.001
125.059	10.93	5-Methylcytosine	1.826	ns	1.021	ns	2.013	<0.001	2.139	0.002
244.069	12.15	Pseudouridine	1.780	0.007	1.058	ns	2.266	<0.001	2.134	<0.001
243.085	12.15	Cytidine *	3.051	0.009	1.812	ns	4.662	<0.001	4.849	<0.001
176.043	16.65	*N*-Carbamoyl-l-aspartate	0.576	<0.001	0.446	<0.001	0.295	<0.001	0.351	<0.001
324.036	15.18	UMP *	0.417	<0.001	0.412	<0.001	0.475	<0.001	0.482	<0.001
404.002	16.59	UDP *	0.218	<0.001	0.347	<0.001	0.413	<0.001	0.350	<0.001
536.044	16.19	UDP-d-xylose	0.593	<0.001	0.699	0.001	1.017	ns	0.745	0.002
580.034	18.96	UDP-glucuronate	0.615	<0.001	0.736	<0.001	0.701	<0.001	0.664	<0.001
566.055	16.31	UDP-glucose *	0.539	<0.001	0.662	<0.001	0.628	<0.001	0.620	<0.001
		**Tryptophan Metabolism**								
175.063	10.37	Indole-3-acetate	1.537	<0.001	1.545	<0.001	0.417	<0.001	1.527	<0.001
236.079	10.37	l-Formylkynurenine	1.507	0.002	1.431	<0.001	0.418	<0.001	1.453	<0.001
191.058	10.37	5-Hydroxyindoleacetate *	1.512	<0.001	1.506	<0.001	0.523	<0.001	1.547	<0.001
208.085	11.15	l-Kynurenine *	1.483	ns	1.114	ns	1.565	0.003	2.087	<0.001
220.085	9.99	5-Hydroxy-l-tryptophan isomer	8.354	<0.001	7.721	<0.001	1.655	0.007	8.798	<0.001
205.074	7.56	Indolelactate	2.315	0.009	1.399	ns	2.267	<0.001	3.333	<0.001
117.058	11.92	Indole *	1.413	0.011	1.216	ns	3.554	<0.001	2.257	<0.001
204.090	11.91	l-Tryptophan *	1.311	ns	1.050	ns	3.194	<0.001	2.019	0.003
		**Miscellaneous**								
146.069	15.31	l-Glutamine *	1.445	0.001	1.105	ns	1.965	<0.001	1.889	<0.001
147.053	14.71	d-Glutamate *	1.035	ns	0.777	0.002	1.259	0.002	1.355	0.003
301.056	14.93	*N*-Acetyl-d-glucosamine 6-phosphate *	0.785	0.006	0.755	<0.001	0.642	<0.001	0.878	ns
103.100	20.65	Choline *	1.277	ns	1.336	0.002	1.268	0.005	1.944	<0.001
141.019	15.91	Ethanolamine phosphate *	0.869	ns	0.987	ns	0.555	<0.001	0.797	0.016
105.043	16.02	l-Serine *	1.646	0.004	1.228	0.030	1.835	<0.001	1.927	<0.001
119.058	14.68	l-Threonine	2.719	<0.001	1.618	0.001	3.190	<0.001	2.821	<0.001
155.070	15.16	l-Histidine *	2.450	<0.001	1.285	0.018	3.175	<0.001	2.737	<0.001
146.106	25.28	l-Lysine *	1.520	0.001	1.254	0.017	1.862	<0.001	1.734	<0.001
384.122	13.99	S-Adenosyl-l-homocysteine *	1.163	ns	1.047	ns	1.580	0.001	1.545	0.002
203.116	11.29	O-Acetylcarnitine *	0.702	0.001	0.688	<0.001	0.644	<0.001	0.689	0.001
175.048	14.53	*N*-Acetyl-l-aspartate *	0.758	0.003	0.628	<0.001	0.797	0.003	0.970	ns
133.038	15.04	l-Aspartate *	1.136	ns	0.888	ns	1.306	0.003	1.618	<0.001
89.048	14.97	l-Alanine *	0.671	0.001	0.731	0.004	1.041	ns	0.720	0.001
226.106	16.03	Carnosine *	1.088	ns	1.080	ns	1.203	0.016	1.201	0.012
132.054	15.48	l-Asparagine *	1.211	ns	0.798	ns	1.310	0.001	1.416	0.001
149.051	11.79	l-Methionine *	1.358	ns	0.910	ns	1.822	<0.001	1.869	<0.001
222.067	17.33	Cystathionine *	0.714	ns	0.833	ns	0.508	<0.001	0.811	ns
181.074	13.24	l-Tyrosine *	1.360	ns	0.989	ns	1.930	<0.001	1.840	0.001
131.095	11.11	l-Leucine *	1.383	0.007	0.962	ns	1.836	<0.001	1.949	<0.001
131.094	11.51	l-Isoleucine *	1.424	ns	0.915	ns	1.838	<0.001	1.852	<0.001
117.079	12.77	l-Valine *	1.739	0.006	1.045	ns	1.901	<0.001	1.876	<0.001

Rt: Retention time (min); LPS: Lipopolysaccharides; *: Matches the analytical standard retention time; ns: Non-significant; n/a: Not applicable.

**Table 3 vaccines-07-00142-t003:** Significantly changed non-polar metabolites in THP-1 cells treated with lipopolysaccharide (LPS), alone or in combination with one of three synthetic forms of honey bee eicosenoids (0.5 µg/mL LPS; 9 µg/mL 11E-OH; 150 µg/mL 11E-ester; 40 µg/mL 11E-acid). Data are compared with those from untreated control cells.

Mass	Rt	Putative Metabolite	LPS/C		11E-OH + LPS/C		11E-Ester + LPS/C		11E-Acid + LPS/C	
			Ratio	*p* Value	Ratio	*p* Value	Ratio	*p* Value	Ratio	*p* Value
	**Fatty Acid and Related Metabolites**									
306.256	19.23	Icosatrienoic acid *	1.963	<0.001	2.976	0.001	7.296	<0.001	2.941	<0.001
328.240	18.11	Docosahexaenoic acid *	1.637	0.007	2.005	0.003	4.187	<0.001	2.336	<0.001
336.303	22.99	Docosadienoic acid *	1.058	ns	1.021	ns	2.301	<0.001	1.027	ns
282.256	19.94	Oleic acid *	1.126	ns	1.189	0.034	2.938	<0.001	1.391	0.003
216.136	4.28	Undecanedioic acid	1.181	0.041	1.463	0.001	9.890	<0.001	1.487	0.003
214.193	14.92	Tridecanoic acid *	1.138	ns	1.102	ns	1.731	0.018	1.323	0.004
334.287	21.67	Docosatrienoic acid *	1.288	0.050	1.346	0.018	3.345	<0.001	1.523	0.006
398.339	24.40	Axillarenic acid *	1.167	ns	1.118	0.023	1.331	0.039	1.190	0.007
366.350	26.47	Tetracosenoic acid *	1.322	0.001	1.632	<0.001	6.416	<0.001	4.085	0.018
230.152	7.29	Dodecanedioic acid	1.071	ns	1.313	ns	23.842	<0.001	1.352	0.023
258.183	12.16	Tetradecanedioic acid	1.342	ns	1.231	ns	23.897	0.009	1.573	0.030
242.225	17.85	Pentadecanoic acid *	1.021	ns	1.084	ns	1.711	0.017	1.233	0.039
248.178	12.32	Hexadecatetraenoic acid	1.205	ns	1.837	ns	2.553	0.017	2.041	0.044
298.287	23.03	Nonadecanoic acid *	1.133	0.001	1.133	ns	1.310	0.001	1.219	<0.001
332.272	20.18	Docosatetraenoic acid *	1.664	0.004	2.549	<0.001	4.261	<0.001	2.348	<0.001
304.240	18.24	Eicosatetraenoic acid *	1.578	0.006	2.874	<0.001	6.998	<0.001	2.985	<0.001
338.319	24.50	Docosenoic acid *	1.155	0.050	1.871	<0.001	22.311	<0.001	8.335	<0.001
394.381	28.24	Hexacosenoic acid *	1.433	<0.001	1.047	ns	1.903	<0.001	1.685	<0.001
364.334	25.02	Tetracosadienoic acid *	0.888	ns	1.409	ns	3.193	<0.001	2.568	<0.001
202.120	4.64	Decanedioic acid	1.134	ns	1.441	0.008	1.944	0.007	1.204	ns
226.193	15.21	Tetradecenoic acid *	0.820	ns	0.941	ns	1.660	0.009	0.963	ns
186.162	11.81	Undecanoic acid *	1.065	ns	0.898	ns	1.864	0.015	1.238	ns
158.131	8.54	Nonanoic acid *	1.084	ns	0.946	ns	1.714	0.019	1.370	ns
280.240	18.26	Linoleate *	1.169	ns	1.052	ns	1.617	<0.001	1.202	ns
172.110	4.54	9-Oxononanoic acid	1.213	ns	1.231	ns	1.497	ns	1.388	0.031
368.220	11.57	Prostaglandin G2	1.358	ns	1.841	0.026	3.300	0.003	2.432	0.008
356.257	14.60	Prostaglandin F1alpha	1.203	ns	1.446	0.021	1.927	<0.001	1.544	0.001
		**Glycerophospholipids**								
638.396	13.80	PA (32:5)	1.549	ns	1.510	ns	2.748	0.021	2.678	ns
847.645	29.88	PC (42:6)	0.963	ns	3.742	<0.001	4.541	<0.001	2.501	0.019
825.530	18.21	PC (40:10)	0.897	0.042	0.753	0.031	0.551	0.002	0.563	<0.001
851.546	18.59	PC (42:11)	0.821	0.016	0.574	<0.001	0.269	<0.001	0.406	<0.001
881.593	21.81	PC (44:10)	1.016	ns	1.743	0.002	2.051	<0.001	1.090	ns
722.545	24.76	PG (33:0)	0.884	ns	0.933	ns	0.857	ns	0.515	0.001
484.280	22.13	Lyso PG (16:0)	1.056	ns	0.876	ns	0.817	0.002	0.942	ns
482.264	7.32	Lyso PG (16:1)	0.833	ns	1.097	ns	0.820	0.050	0.942	ns
572.296	7.29	Lyso PI (16:0)	1.028	ns	1.137	0.027	1.268	0.001	1.200	0.001
598.312	7.95	Lyso PI (18:1)	1.031	ns	1.241	0.007	1.563	<0.001	1.303	0.004
622.312	7.71	Lyso PI (20:3)	1.119	ns	1.302	0.020	1.154	ns	1.255	0.031
620.296	7.04	lyso PI (20:4)	1.152	ns	1.187	0.045	1.891	<0.001	1.770	<0.001
495.260	6.38	lyso PS (16:1)	1.015	ns	0.885	ns	0.730	0.004	0.818	0.036
771.505	24.05	PS (35:3)	0.647	0.001	0.434	<0.001	0.315	<0.001	0.272	<0.001
497.275	7.72	Lyso PS (16:0)	1.033	ns	1.139	ns	0.801	0.011	1.036	ns
517.244	12.63	Lyso PS 18:4	1.689	0.040	1.922	<0.001	2.208	0.020	1.390	ns
691.441	12.63	PS (29:1)	1.380	ns	2.276	0.038	2.380	ns	2.448	ns

Rt: Retention time (min); LPS: Lipopolysaccharides; *: Matches the analytical standard retention time; ns: Non-significant; n/a: Not applicable.

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
