# Peer review of "Metabolomic Profiling of the Immune Stimulatory Effect of Eicosenoids on PMA-Differentiated THP-1 Cells"

_vaccines, 2019, doi:10.3390/vaccines7040142_

Round 1

Reviewer 1 Report

Manuscript ID: vaccines-599241

Metabolomic Profiling of the Immune Stimulatory Effect of Eicosenoids on PMA-Differentiated THP-1 Cells

The manuscript focuses on the investigation of the honey bee venom constituent
(Z)-11-eicosenol-a and two synthetical derivates as possible new vaccine adjuvants. The in vitro studies were performed with PMA-differentiated macrophage-like THP-1 cells by measuring the production of selected cytokines following stimulation with eicosenoids, either alone or in combination with LPS. The authors unveiled eicosenoid-specific cytokine profiles suggesting a complex immunologic impact. Moreover, in these cells they found eicosenoid-specific profiles of polar as well as lipophilic metabolites via robust untargeted LC-MS-based metabolomics. The authors linked their comprehensive results to immunological relevant pathways, and finally concluded that eicosenoids may serve as novel vaccine adjuvants in case of further supportive data.

Specific comments for revision:

Major comments:

The text of the manuscript has to be revised, especially the “Results” part (typos, wording and style!). Results 3.1 to 3.5: Controls are missing. - To evaluate the data accurately the authors should add data for the used solvent (DMSO, I assume). If there were no differences between the untreated and the solvent control, a simple commentary would be sufficient.

Minor comments:

Methods 2.6, line 181: I assume it should be worded: “Extraction of the metabolites…” The point “Statistical Analysis” (2.8) is listed, however, no statistical tests are defined. The authors have to declare which statistical test(s) they used. Methods 2.3 & Results 3.1: The controls comprise DMSO, but the authors don’t declare why. I assume DMSO is the solvent for LPS and the eicosenols. This has to be clarified. In line 161, certainly the authors wanted to say “…and the cell viability…” Results 3.2line 212: It would be better to say: “The levels of secreted TNF-a by PMA-diff. …”In addition, even if significant, it is at least questionable whether such as small difference is biologically relevant. Fig. 2: It is hard to belief that the difference in TNF-a secretion between LPS and LPS+11E-acid is significant. The authors should mention the statistical test used. line 211: For clarity, the used method (ELISA, I assume) should be mentioned. Results 3.4, last sentence: The remark “the measured IL-6 was not signif. different from the background” is misleading (no test is presented and a calculation with “<2.0” is not possible). For example, the authors could say: “Unstimulated macrophage-like THP-1 cells did not produce any IL-6. The same was true for cells stimulated with E’s alone.” Figures 6, 7 and 8: Comments have to be deleted. (Same is true for the last line). Tables 2 and 3 would be easier to read if the table header would resume on every page of those large tables. Discussion, line 301: I suggest the following wording: “… derivates to induce specific immunological functions of THP-1 macrophages…” Line 323: “… was observed for the secretion of TNF-a…” Lines 371f and 382f: The authors should provide at least one reference for each statement. Figure 9, figure legend: the most significant level has to be represented by 3 asterisks (***) The Reference List has to be revised due to typos and incompleteness. For instance, in Ref. 17, information concerning the Journal, etc. is missing.

Author Response

Major comments:

The text of the manuscript has to be revised; especially the “Results” part (typos, wording and style!). Results 3.1 to 3.5: Controls are missing. - To evaluate the data accurately the authors should add data for the used solvent (DMSO, I assume). If there were no differences between the untreated and the solvent control, a simple commentary would be sufficient.

Response: DMSO is the solvent used to dissolve the sample at the beginning (10 mg/mL, stock solution), after that any serial dilutions for the samples have been made by using fresh media (to reach1.2 to 150 µg/mL of samples and 1.5 % final concentration of DMSO). In IC50, DMSO was treated exactly as the samples and final concentration of DMSO (1.5%) used, which assure that DMSO has no effect on cell viability, so no further need to test DMSO alone in the cytokines experiments (results 3.2 to 3.5).

The final chosen concentration in Table 1 were chosen with >98% cell viability for the samples and no effect of DMSO observed.

Note: LPS dissolved directly to media.

However, sentence added in the result 3.1 clarifying this point and the percent calculated for DMSO was added in the method 2.3.

Minor comments:

Methods 2.6, line 181: I assume it should be worded: “Extraction of the metabolites…”

Response:The method has been amended.

The point “Statistical Analysis” (2.8) is listed, however, no statistical tests are defined. The authors have to declare which statistical test(s) they used.

Response:The following sentence added “Univariate comparisons were performed using Microsoft Excel and paired t-tests between treated and control cells and differences were considered significant at p < 0.05.”

Methods 2.3 & Results 3.1: The controls comprise DMSO, but the authors don’t declare why. I assume DMSO is the solvent for LPS and the eicosenols. This has to be clarified.

Response:See above.

In line 161, certainly the authors wanted to say “…and the cell viability…”

Response:Yes the cell viability.

Results 3.2line 212: It would be better to say: “The levels of secreted TNF-a by PMA-diff. …”In addition, even if significant, it is at least questionable whether such as small difference is biologically relevant. Fig. 2: It is hard to belief that the difference in TNF-a secretion between LPS and LPS+11E-acid is significant. The authors should mention the statistical test used. line 211: For clarity, the used method (ELISA, I assume) should be mentioned.

Response: I think we state that there is no much additional effect of the eicosanoids over that of LPS on TNF. Amended to include “The levels of secreted TNF-a by PMA-diff. …”

The statistical test added to the method 2.8 and the difference between LPS and 11E-acid+LPS was significant.

“Using ELISA” added.

Results 3.4, last sentence: The remark “the measured IL-6 was not signif. different from the background” is misleading (no test is presented and a calculation with “<2.0” is not possible). For example, the authors could say: “Unstimulated macrophage-like THP-1 cells did not produce any IL-6. The same was true for cells stimulated with E’s alone.”

Response: Sentence amended as suggested in result 3.4

Figures 6, 7 and 8: Comments have to be deleted. (Same is true for the last line).

Response: Not clear.

Tables 2 and 3 would be easier to read if the table header would resume on every page of those large tables.

Response: The table header added as suggested.

Discussion, line 301: I suggest the following wording: “… derivates to induce specific immunological functions of THP-1 macrophages…” Line 323: “… was observed for the secretion of TNF-a…”

Response:The discussion amended as suggested.

Lines 371f and 382f: The authors should provide at least one reference for each statement.

Response: References added.

Figure 9, figure legend: the most significant level has to be represented by 3 asterisks (***)

Response: Legend amended.

The Reference List has to be revised due to typos and incompleteness. For instance, in Ref. 17, information concerning the Journal, etc. is missing. 

Response: The Reference List revised.

Reviewer 2 Report

In this paper, the effect of (Z)-11-eicosenol (that is present in honey bee venom),  and its derivatives methyl cis-11-eicosenoate and cis-11-eicosenoic acid, was evaluated in order to confirm their possible use as vaccine adjuvants. They were evaluated alone or in combination with lipopolysaccharide (LPS), by examining the secretion of tumour necrosis factor-α (TNF-α) and the cytokines interleukin-1β (IL-1β), IL-6 and IL-10 by THP-1 macrophages. From metabolic profiling, the authors studied the effect on the immune response.

In my opinion, the paper should be corrected as follow:

The abstract must contain less technical information and must underline the results of the paper.

The introduction miss the point of the importance of honey bee venom derivatives as adjuvant in vaccines at the end, please stress this point. At the beginning of the introduction the third phrase should be the first one.

In materials and methods, NMR assignment is referred to the number of the atoms, please add a figure of the molecules in the main text.

In paragraph 3.6 (Effect of eicosenoid compounds on polar THP-1 cell metabolites), the techniques should be better described, underlining the information expected by principle component analysis (PCA) and Orthogonal partial least squares discriminant analysis (OPLS-DA).

The discussion is confused and should be made more clear.

The conclusions are missed, since it is not clear if (Z)-11-eicosenol, methyl cis-11-eicosenoate and cis-11-eicosenoic acid, can be used as immune-modulating agents and as vaccine adjuvants better than that actually used.

Figures and Table reported in SI should be cited in the main text.

Author Response

In my opinion, the paper should be corrected as follow:

The abstract must contain less technical information and must underline the results of the paper.

Response: Abstract amended.

The introduction miss the point of the importance of honey bee venom derivatives as adjuvant in vaccines at the end, please stress this point. At the beginning of the introduction the third phrase should be the first one.

Response:The introduction amended as suggested.

In materials and methods, NMR assignment is referred to the number of the atoms, please add a figure of the molecules in the main text.

Response: The figure of the chemical structure of the molecules added in the main text as suggested.

In paragraph 3.6 (Effect of eicosenoid compounds on polar THP-1 cell metabolites), the techniques should be better described, underlining the information expected by principle component analysis (PCA) and Orthogonal partial least squares discriminant analysis (OPLS-DA).

Response: For PCA: It is usually used to show the absence of the outliers during the samples run in LC-MS instruments and the cluster of the quality control samples (pool), which describe the stability and precision of the run, which is already described in the result 3.6.

For OPLS-DA: it is fully described which provide a full group separation and discrimination, and it was valid.  These indicate a unique metabolite profiling that distinguish the groups from one another.

The discussion is confused and should be made more clear.

Response: We have re-read the discussion and made some changes. The metabolic changes are complex and we have done our best to describe and rationalise them.

The conclusions are missed, since it is not clear if (Z)-11-eicosenol, methyl cis-11-eicosenoate and cis-11-eicosenoic acid, can be used as immune-modulating agents and as vaccine adjuvants better than that actually used.

Response: An additional sentence has been added in the conclusion section.

Figures and Table reported in SI should be cited in the main text.

Response: Figures cited and method 2.8 and tables cited as suggested.

This manuscript is a resubmission of an earlier submission. The following is a list of the peer review reports and author responses from that submission.